# Arginine and Endothelial Function

**DOI:** 10.3390/biomedicines8080277

**Published:** 2020-08-06

**Authors:** Jessica Gambardella, Wafiq Khondkar, Marco Bruno Morelli, Xujun Wang, Gaetano Santulli, Valentina Trimarco

**Affiliations:** 1Department of Medicine (Division of Cardiology), Wilf Family Cardiovascular Research Institute, Albert Einstein College of Medicine—Montefiore University Hospital, New York City, NY 10461, USA; jessica.gambardella@einsteinmed.org (J.G.); wakhonda22@herricksk12.org (W.K.); marco.morelli@einsteinmed.org (M.B.M.); xujun.wang@einsteinmed.org (X.W.); 2Department of Molecular Pharmacology, Fleischer Institute for Diabetes and Metabolism, Albert Einstein College of Medicine, New York City, NY 10461, USA; 3Department of Advanced Biomedical Sciences, “Federico II” University, 80131 Naples, Italy; 4International Translational Research and Medical Education (ITME), 80100 Naples, Italy; 5Department of Neuroscience, “Federico II” University, 80131 Naples, Italy; valentina.trimarco@unina.it

**Keywords:** ADMA, arginine, arginine paradox, BH4, blood pressure, COVID-19, dietary supplements, endothelial dysfunction, endothelium, eNOS uncoupling, heart failure, hypertension, L-arginine, myocardial infarction, NADPH, nitric oxide, oxidative stress, peripheral artery disease

## Abstract

Arginine (L-arginine), is an amino acid involved in a number of biological processes, including the biosynthesis of proteins, host immune response, urea cycle, and nitric oxide production. In this systematic review, we focus on the functional role of arginine in the regulation of endothelial function and vascular tone. Both clinical and preclinical studies are examined, analyzing the effects of arginine supplementation in hypertension, ischemic heart disease, aging, peripheral artery disease, and diabetes mellitus.

## 1. Pleiotropic Effects of Arginine

L-arginine, hereinafter referred to as arginine, is a semi-essential or conditionally essential amino acid, since it can be synthetized by healthy individuals but not by preterm infants [1]. From a chemical point of view, arginine is a 2-amino-5-guanidinopentanoic acid (Figure 1). Its name derives from the Greek word ἄργυρος (silver), indicating the color of arginine nitrate crystals.

Arginine is involved in a number of biological processes, it is the substrate for a series of reactions leading to the synthesis of other amino acids, and it is a substrate for two enzymes, namely nitric oxide (NO) synthase (NOS) and arginase, which are fundamental for the generation of NO and urea, respectively. Arginine is known to act as a substrate for NO production by endothelial cells, thus regulating vascular tone and, overall, cardiovascular homeostasis [2]. NO is synthesized from arginine by the enzyme NOS in a reaction that involves the transfer of electrons from nicotinamide adenine dinucleotide phosphate (NADPH)—via the flavin adenine dinucleotide (FAD) and flavin mononucleotide (FMN) in the C-terminal reductase domain [3,4]—to the heme in the N-terminal oxygenase domain, where the substrate arginine is oxidized to citrulline and NO [5,6], as shown in Figure 1. Arginine is also implicated in T-cell proliferation and host immune responses, as well as in creatine and collagen synthesis [7,8,9,10,11].

There are three isoforms of NOS, two of which—endothelial (eNOS) [12,13] and neuronal (nNOS) [14,15,16]—are constitutively expressed, while the third one, inducible NOS (iNOS) [17,18,19], is expressed in response to cytokines and is related to the inflammatory response [6,20]. NO generation occurs in two steps: first, NOS hydroxylates arginine to *N*^ω^-hydroxy-arginine (which remains largely bound to the enzyme); in a second step, NOS oxidizes *N*^ω^-hydroxy-arginine to citrulline and NO [21,22,23,24,25,26,27,28,29].

In normal conditions, NOS catalyzes the transformation of arginine, O_2_, and NADPH-derived electrons to NO and citrulline (Figure 1). However, in the presence of pathologic conditions like atherosclerosis and diabetes, the NOS function is altered, and the enzyme catalyzes the reduction of O_2_ to superoxide (O_2_^−^), a phenomenon that is generally referred to as “NOS uncoupling” [30,31,32,33,34,35,36,37,38,39,40,41], and has been linked to a limited bioavailability of tetrahydrobiopterin (BH4, also known as sapropterin) [42,43,44,45,46,47]. Indeed, the donation of an electron by BH4 to produce a transient BH4^•+^ radical is required for the oxidation of arginine to citrulline and the associated formation of a ferrous iron–NO complex at the NOS heme catalytic center [48,49,50,51]. BH4 is synthesized from guanosine triphosphate (GTP) by GTP cyclohydrolase I (GTPCH) and recycled from 7,8-dihydrobiopterin (BH2) by dihydrofolate reductase (Figure 2). Of note, NOS is inhibited by arginine analogs that are substituted at the guanidino nitrogen atom, like NG-monomethyl-arginine or NG-nitro-arginine [52,53,54,55,56,57,58].

As mentioned above, in the urea cycle arginine is converted by arginase, a manganese metalloenzyme, in ornithine and urea; this cycle is crucial not only for allowing urea excretion, but also for producing bicarbonate, which is critical for maintaining acid/base homeostasis [59,60,61,62,63]. Arginase exists in two distinct isoforms, arginase I and II, that share ~60% sequence homology; arginase I is a cytosolic enzyme mainly localized in the liver, whereas arginase II is a mitochondrial enzyme with a wide distribution and is expressed in the kidney, prostate, gastrointestinal tract, and the vasculature [64,65,66,67].

The enzyme arginase is a key modulator of NO production by competing for arginine: in other words, NO generation is dependent on the relative expression and activities of arginase and NOS. More specifically, increased arginase activity may lead to a decreased bioavailability of arginine for NOS, thereby diminishing NO production. This mechanism has emerged as an essential factor underlying impaired endothelial functions [68,69]. Specifically, an increased arginase activity has been associated with endothelial dysfunction in a number of experimental models of hypertension, atherosclerosis, diabetes, and aging [70,71,72,73,74,75,76,77,78,79,80,81,82,83,84,85,86,87,88,89,90,91,92].

## 2. Arginine and NO Production in Physiological Conditions: The Arginine Paradox

Indeed, endothelial dysfunction is a leading cause of several pathological conditions affecting the cardiovascular system, including hypertension, atherosclerosis, diabetes, and atherothrombosis [46,93,94,95,96,97,98,99,100,101,102,103,104,105,106,107,108,109,110,111,112,113,114,115,116,117,118,119]. Moreover, in April 2020, we were the first group to show that the systemic manifestations observed in coronavirus disease (COVID-19), caused by the severe acute respiratory syndrome coronavirus 2 (SARS-CoV-2), could be explained by endothelial dysfunction [120]. Indeed, alterations in endothelial function have been linked to hypertension, diabetes, thromboembolism, and kidney failure, all featured, to different extents, in COVID-19 patients [121,122,123]. Other investigators have later confirmed our view [124,125,126,127,128,129,130,131,132,133]. On these grounds, based on the positive effects of arginine on endothelial function, we can also speculate that arginine supplementation could be helpful, while not being harmful, for contrasting endothelial dysfunction in COVID-19 patients.

An increasing interest in the potential therapeutic effects of arginine supplementation, especially in cardiovascular disorders, has recently emerged. An impaired NO synthesis is considered a main feature of a dysfunctional endothelium [107,134,135,136]; however, several studies suggest that arginine supplementation in healthy subjects does not lead to a significant increase in NO production [11,137,138,139,140]. For instance, the daily administration of arginine for 1 week did not affect the serum concentration of two established indicators of NO production, namely NO_2_^−^ and NO_3_^−^, in twelve healthy subjects [138]. In another study, 20 healthy subjects received daily arginine supplementation in both sustained-release or immediate-release form; despite the significant increase in the plasma arginine concentration, which proved the effectiveness of the administration protocol, the authors did not observe significant differences in urinary extraction of nitrate [141].

One reason for the absence of significant results in normal conditions could be that the NO synthesis machinery seems to be saturated by the endogenous arginine. Indeed, the Michaelis–Menten constant (K_m_) of NO synthase is in the micromolar range, specifically 2.9 μmol/L, as demonstrated by Bredt and colleagues [142]. Arginine plasma levels measured in healthy humans are 15–30-fold higher than this K_m_, thereby making the levels of the substrate a non-limiting factor in the enzymatic reaction leading to NO production. Despite such a biochemical ratio, which in fact makes the enzyme physiologically saturated, various studies are also showing beneficial effects of arginine supplementation in healthy subjects. For instance, arginine supplementation has been tested in athletes, as vasodilation favors muscle perfusion and nutrient/oxygen delivery during exercise, enhancing muscle strength and recovery [143]. Controversial results come from these studies, sometimes yielding no effects of arginine supplementation on muscle performance, and sometimes demonstrating a significant improvement in exercise capability [137,144,145,146,147,148].

The phenomenon known as “arginine paradox” is born from this scenario, and indicates that we were losing part of the story concerning the alternative ways by which arginine can act on endothelial NO production. The arginine paradox refers to the fact that despite intracellular physiological concentrations of arginine being several hundred micromoles per liter, thereby exceeding the K_m_ of eNOS, the acute provision of exogenous arginine still increases NO production [149,150,151]. 

One of the mechanisms that may help explain the arginine paradox comes from the discovery of asymmetric dimethylarginine (ADMA), an endogenous inhibitor of NOS [152,153,154,155]. Given its own structure similar to arginine, ADMA is a direct competitor for NOS binding. Moreover, both ADMA and arginine are both transported into the cell via the cationic amino acid transporter (CAT, also known as “y^+^ system”), a high-affinity, Na^+^-independent transporter of the basic amino acids [156,157], and therefore also compete with each other on this level (Figure 2). Since ADMA competes with arginine for NOS and for cell transport, the bioavailability of NO depends on the balance between the two [158]. Plasma levels of ADMA increase during hypertension, hypercholesterolemia, diabetes, and atherosclerosis [95,159,160,161,162,163,164,165,166,167,168,169,170]. Hence, despite the range of endogenous arginine levels, they could still be sufficient to guarantee eNOS saturation, and so the arginine/ADMA ratio would be reduced, resulting in a net inhibition of NO production [171,172,173]. 

The arginine/ADMA ratio is widely considered to be an important indicator of NO bioavailability as well as of the risk of formation of atherosclerotic plaques [174]. The ratio has been shown to be a better predictor for all-cause mortality compared to ADMA alone [174,175]. Similarly, although plasma ADMA levels were a significant predictor of all-cause mortality in an elderly population, the effect disappeared in subjects with higher arginine levels [176], and the arginine/ADMA ratio (but not ADMA alone) was a significant risk factor for microangiopathy-related cerebral damage in an elderly population [177].

Arginine supplementation can equilibrate the arginine/ADMA ratio, recovering the production of NO. In other terms, the increased arginine availability, resulting from supplementation, competes with ADMA in binding eNOS (Figure 2). This interesting mechanism sheds light on the effectiveness of the increased arginine availability, implicating further therapeutic options for arginine supplementation. Furthermore, this phenomenon can explain some conflicting results about arginine supplementation studies, as ADMA levels should be considered in the study populations. Specifically, cardiovascular patients with increased ADMA plasma levels could be the best target of arginine supplementation.

Another potential explanation of the arginine paradox may be that arginine could be compartmentalized in the cytoplasm, and local concentrations in the vicinity of NOS may be lower than expected from arginine levels in whole-cell homogenates [178].

## 3. Impaired NO Production as a Mechanism of Endothelial Dysfunction and Arginine Intervention

The major determinants of cardiovascular risk, including dyslipidemia, glucose intolerance, smoking, hypercholesterolemia, and aging, have a direct impact on the endothelium [179,180,181]. Exposing the vasculature to these conditions induces endothelial dysfunction and alterations as an early phenomenon, able to evolve and contribute to the progression towards clinically relevant disorders like hypertension, atherosclerosis, and diabetes mellitus. Hence, the endothelium plays a key role in cardiovascular physiology and pathophysiology [182,183,184,185,186,187,188,189,190,191,192,193,194]. Fervent research has been conducted in recent years in order to understand the underlying mechanisms and identify therapeutic strategies to prevent or counteract endothelial dysfunction.

The ability of the endothelium to regulate vascular homeostasis is largely dependent on NO production, making endothelial vasodilator failure the main sign of endothelial dysfunction and a hot point to be targeted. The impaired endothelial NO availability in perturbed vasculature can be attributable to a diminished synthesis of NO or, indirectly, to an increased ROS production, which inactivates the NO source [195,196]. In addition to counteracting oxidative stress, the stimulation of NO synthesis represents an alternative and a potentially effective approach [197,198], for instance, by providing further substrates to NO synthase. Theoretically, arginine supplementation meets these needs, and thus, it has been tested in many cardiovascular disorders as a potential therapeutic strategy [199]. However, human studies on arginine supplementation have often been a source of debate. Indeed, in healthy subjects as well as in patients suffering from cardiovascular disorders, levels of plasma arginine range from ~45 to ~100 μmol/L [137,200,201,202], significantly higher than the eNOS K_m_ of 2.9 μmol/L [203]. Endocrine mechanisms may also contribute to vasodilation induced by arginine. Indeed, arginine stimulates the release of both insulin [204,205,206] and glucagon [207] from pancreatic islets of Langerhans. Interestingly, an intravenous infusion of arginine has been shown to induce vasodilation and insulin release in healthy humans, but when insulin secretion was blocked by octreotide co-infusion, no vasodilation occurred, whereas vasodilation was restored by insulin co-administration [208]. Since high intravenous doses of arginine (30 g) have also been shown to induce growth hormones (GHs), and secretion [209], the vasodilation induced by arginine could also be mediated by GHs via a signaling pathway that includes insulin-like growth factor-1 [210,211].

Substantial data indicate that endothelial dysfunction is highly prevalent in elderly individuals [212,213]. Endothelial dysfunction has also been implicated in age-associated declines in cognitive function, physical function, as well as in the pathogenesis of stroke, erectile dysfunction, and renal dysfunction. Clinical trials testing the effects of arginine in aging-induced endothelial dysfunction have yielded controversial results. An acute intravenous infusion of arginine (1 g/min for 30 min) had no effect on endothelial-dependent vasodilation in healthy older individuals [214]. Similarly, the intravenous infusion of arginine induced a significant increase in the renal plasma flow, glomerular filtration rate, natriuresis, and kaliuresis, in young but not in aged hypertensives [215]. Another study conducted in healthy postmenopausal women taking 9 g of arginine per day for 1 month confirmed that plasma arginine increased without a concomitant significant change in flow-mediated dilation [216]. On the contrary, in a prospective, double-blind, randomized crossover trial in 12 healthy, old participants (age 73.8 ± 2.7 years), chronic arginine supplementation (16 g/day for 2 weeks) markedly increased their plasma levels of arginine (114.9 ± 11.6 vs. 57.4 ± 5.0 mM) and significantly improved endothelial-dependent vasodilation [217]. 

## 4. Arginine Supplementation in Hypertension

The majority of studies in animal models supports a beneficial effect of arginine supplementation in hypertension, especially in the presence of salt-sensitive hypertension. For instance, both oral [218,219,220] and intraperitoneal [221,222] arginine administration in Dahl salt-sensitive (DSS) rats was shown to prevent the increase in blood pressure induced by a high salt diet. However, arginine was not effective in DSS pretreated with high salt for three weeks [218], suggesting that arginine is able to prevent and counteract hypertension when it is in the early stages, but probably not when some changes and pathological remodeling have already occurred.

The outcome of arginine supplementation could also depend on the method of administration. For instance, renal medullary interstitial infusion of arginine prevents the increase in blood pressure in high salt-treated rats, while the intravenous dose necessary to obtain a similar increase in plasma arginine does not affect blood pressure [223]. A rat model of type 1 diabetes mellitus shows an important reduction in blood pressure after 4 weeks of oral arginine treatment [224]; oral arginine administration prevents fructose-induced hypertension [225]. Oral arginine administration does not correct hypertension in spontaneously hypertensive rats, although markedly reduces renal damage [226].

Although the beneficial effect of arginine supplementation in hypertension appears to be largely attributable to its impact on NO synthesis, arginine has also been shown to have antioxidant properties, thus affecting the activity of redox-sensitive proteins and lowering blood pressure [227,228,229,230,231,232,233,234]. Indeed, supplementation with 3 g/day arginine for two months increases the serum total antioxidant capacity in obese patients with prediabetes [235]; of note, in vitro experiments performed in endothelial cells have revealed that arginine reduces superoxide release and the cell-mediated breakdown of NO [236].

In the clinical scenario, the oral administration of arginine acutely improves endothelium-dependent, flow-mediated dilatation of the brachial artery in patients with essential hypertension [237]; however, the long-term effects of arginine were not investigated in this study [237]. In a Japanese population, the acute intravenous infusion of arginine (500 mg/kg for 30 min) is able to decrease arterial pressure of both salt-sensitive and salt-insensitive patients [238]. In a similar study, conducted on African-Americans, the same amount of arginine administration reduces blood pressure with a greater effect in the salt-sensitive population [239]. Interestingly, in hypertensive patients in which the control of blood pressure with angiotensin converting enzyme (ACE)-inhibitors and diuretics for three months was unsuccessful, the addition of oral arginine (6 g/day) was effective in reducing both systolic and diastolic blood pressure levels [240]. Unfortunately, many of the findings on the effects of arginine supplementation in hypertension derive from small clinical studies and, despite the promising efficacy, further investigations are needed, especially large, randomized, and controlled trials. The ability to modulate the renin-angiotensin-aldosterone system (RAAS) is another mechanism by which arginine can regulate blood pressure: specifically, arginine inhibits ACE activity, reducing angiotensin II production and its effects on vascular tone [241].

## 5. Arginine Supplementation in Ischemic Heart Disease and Peripheral Artery Disease

Alongside the preservation of endothelial-dependent vasodilation, the enhanced bioavailability of NO reduces the activation of pro-inflammatory genes and the expression of endothelial adhesion molecules [242]. These events strongly regulate the development and the fate of atherosclerosis [243,244,245]. For these reasons, it is not surprising that arginine has a powerful effect on atherogenesis and its evolution. In particular, preclinical investigations have shown that chronic arginine administration in LDL-receptor KO mice significantly reduces the extension of atherosclerotic plaques [246]. Similarly, arginine supplementation in humans reverses the increased monocyte–endothelial adhesion, mirrored by a normalization of platelet aggregation [247]. These effects make arginine a promising drug for disorders like coronary artery disease (CAD), heart failure, and peripheral artery disease (PAD).

In 1997, two important studies investigating the effects of arginine in CAD were published [248,249]. In a placebo-controlled study, Adams and collaborators showed that oral administration of arginine (21 g/day for 3 days) significantly improved the vasodilatory response of the brachial artery in premature CAD [248]. A double-blind placebo-controlled study conducted on 22 patients with stable angina pectoris revealed that the administration of arginine was able to improve their exercise capacity in just 3 days [249]. The following year, a clinical study confirmed the beneficial effects of long-term arginine supplementation (9 g for 6 months), showing significantly enhanced vascular responses to acetylcholine in patients with coronary atherosclerosis [250]. Preclinical studies were consistent with these findings. For instance, oral administration of arginine reduced the intimal hyperplasia in balloon-injured carotid arteries in spontaneously hypertensive rats [251]. This first encouraging evidence prompted further investigations about arginine’s effects on CAD. Again, arginine treatment for 4 weeks preserved endothelial function in CAD patients, markedly reducing LDL oxidation [252]. Another study highlighted the method of administration as a major determinant of the efficacy of high dose arginine supplementation: intra-arterial infusion, but not oral administration, was able to improve endothelial-dependent vasodilation in patients with stable angina pectoris [253].

The therapeutic potential of arginine has been also investigated in heart failure [254,255,256,257,258] and ischemia-reperfusion injury [259,260,261], often yielding controversial results. Endothelium-dependent vasodilation in response to acetylcholine and ischemic vasodilation during reactive hyperemia is attenuated in the forearm of patients with heart failure [262]. In a seminal paper, Hirooka and collaborators demonstrated that the intra-arterial infusion of arginine was effective in reversing the blunted endothelium-dependent vasodilation observed in heart failure [263]. Moreover, oral arginine supplementation (6 g twice a day for 6 weeks) enhanced endurance exercise tolerance in heart failure patients, an important determinant of daily-life activity in patients with chronic stable heart failure [264]. In line with these results, a clinical study carried out in 21 patients with class II/III heart failure (New York Heart Association, NYHA) established that improved endothelial function following exercise training is associated with increased arginine transport [265]. However, another investigation in 20 patients with NYHA class III/IV heart failure demonstrated that responses to acetylcholine and sodium nitroprusside determined using forearm plethysmography were not affected by arginine (20 g/day every day for 28 days), although the actual levels of arginine in the blood were not measured [266]. Exogenous arginine (3 g three times a day for 6 months) administered to patients after an acute myocardial infarction did not improve vascular stiffness measurements or ejection fractions; this clinical trial had to be interrupted due to excess mortality in the treated patients [267].

The improvement in peripheral circulation is critical in patients with PAD, as in severe cases the extensive damage of leg tissues can result in gangrene and amputation [268,269,270]. Intravenous arginine administration to PAD patients is able to increase the calf blood flow and walking distance [271]. Similarly, an acute intravenous arginine infusion (30 g in 60 min) improves NO production and blood flow of the femoral artery in PAD patients [272]. The oral consumption of arginine for 2 weeks is able to increase the pain-free walking distance, improving the quality of life of patients with hypercholesterolemia [273]. Nevertheless, if the short-term arginine administration seems to be effective in treating PAD, the results on long-term administration are less consistent. A randomized clinical trial testing the long-term (6 months) effects of arginine supplementation was conducted on 133 subjects. Despite an increase in plasma levels of arginine, the study revealed no significant effect of arginine treatment on NO-dependent vasodilation, as well as on the relative functional phenotype of PAD patients [274].

## 6. Arginine Supplementation in Diabetes Mellitus

Given the fundamental pathogenic role of endothelial dysfunction in diabetes and its complications [275,276], the therapeutic use of arginine supplementation has been tested. In addition to the direct impact of arginine on endothelial vasodilator capacity, a crosstalk with the insulin pathway has been suggested [150,277]. In particular, as mentioned above, arginine can induce the release of insulin from pancreatic beta cells [204,205,206]. On the other hand, insulin is able to reduce ADMA concentrations [278] and to stimulate the secretion of arginine [279,280]. The stimulation of insulin receptors induces NO release, producing an insulin-dependent vasodilation [281,282,283,284,285]. Of note, such a protective effect of insulin on arginine mobility and endothelial NO production is compromised in diabetes [286]. Henceforth, diabetic patients could be an optimal target population for arginine supplementation.

Preclinical studies corroborate this theory: in diabetic rats, the oral administration of arginine reverses endothelial dysfunction [287], restoring endothelium-dependent relaxation and decreasing oxidative stress [224]. Arginine administration in tap water (free base, 50 mg/kg/day) for 4 months has been shown to reduce both cardiac [288] and renal [289] fibrosis in *db*/*db* mice, by the interaction of arginine with reactive carbonyl residues of glycosylation adducts of collagen, thereby inhibiting glucose-mediated abnormal cross-linking of collagenous structures. These results were later confirmed in a clinical setting, showing that 2 g of arginine free base administered orally as two daily doses of 1 g each reduced the lipid peroxidation product malondialdehyde in diabetic patients [290].

Clinical studies confirmed the reduction in blood pressure, platelet aggregation, and hemodynamic function in diabetic patients treated with intravenous arginine [291]. While in healthy subjects arginine treatment does not seem to affect insulin receptor sensitivity or density [292], in conditions of insulin resistance, arginine improves insulin sensitivity; indeed, the intravenous injection of arginine in obese or type 2 diabetic patients stimulates insulin responsiveness, restoring insulin-dependent vasodilation [151,293]. Similarly, the oral administration of arginine improves hepatic and peripheral insulin sensitivity in a cGMP dependent fashion [294]. A prospective, crossover clinical trial conducted in mildly hypertensive type 2 diabetic patients revealed a significant decrease in blood pressure in response to arginine, occurring two hours after the oral administration; the effect of lowering blood pressure was associated with increased plasma levels of citrulline, whereas no significant changes in insulin levels were detected, suggesting that the observed phenotype was dependent on arginine-induced NO synthesis [295].

Overall, the mentioned studies substantiate the use of arginine in the diabetic population, at least as a prophylactic treatment able to prevent cardiovascular complications of diabetes. One potential limitation for the use of arginine is the risk of reaction with precursors of advanced glycosylated products [296], which are particularly abundant in diabetes. Since the addition of methylglyoxal (abundant in diabetic patients [297]) to arginine has been shown in vitro to produce potent superoxide radicals in a dose-dependent manner [298], arginine supplementation has been suggested to be combined with antioxidants. A double-blind study on 24 diabetic patients verified this assumption evaluating the combination of *N*-acetylcysteine and arginine oral treatments: the combined treatment was able to reduce systolic and diastolic blood pressure, total cholesterol, C-reactive proteins, vascular adhesion molecules, and improved the intima-media thickness during endothelial post-ischemic vasodilation [299]. This last evidence indicates that the combination of arginine with an antioxidant agent should be potentially effective and well-tolerated.

## 7. Conclusions and Perspective: Arginine as a Therapeutic Tool

Overall, data available in the literature support and encourage the use of arginine supplementation in cardiovascular disorders, especially in preventing the evolution of hypertension and atherosclerosis. One limitation of using arginine supplementation remains the selection of the optimal target population. In this sense, we believe that ADMA levels could be very useful in selecting the target population, and patients with increased ADMA/arginine ratios are probably the most suitable population, in which arginine supplementation can actually be effective. Another limitation about arginine use concerns its dose. Indeed, available studies suggest a number of different doses, sometimes effective, sometimes not. For instance, the acute oral administration of arginine (9 g/day) has been shown to be not successful in inducing an effective NO production [216]. Instead, chronic administration of oral arginine (e.g., vials containing arginine salts-free 1.66 g/20 mL), has been shown to favor the utilization of arginine for NO synthesis [300], and we have data showing that oral arginine (3 g/day of Bioarginina^®^, Farmaceutici Damor, 2 vials/day) improves endothelial function in hypertensive patients via the regulation of non-coding RNAs (Gambardella et al., personal communication). Large, prospective randomized clinical trials are needed to better define the target population for arginine supplementation, alongside with correct dosage definitions. To date, a dose of ~3 g/day of arginine (e.g., Bioarginina^®^, 2 vials/day) seems to be effective in favoring the utilization of arginine for NO synthesis, without toxic effects.

## Figures and Tables

**Figure 1 biomedicines-08-00277-f001:**
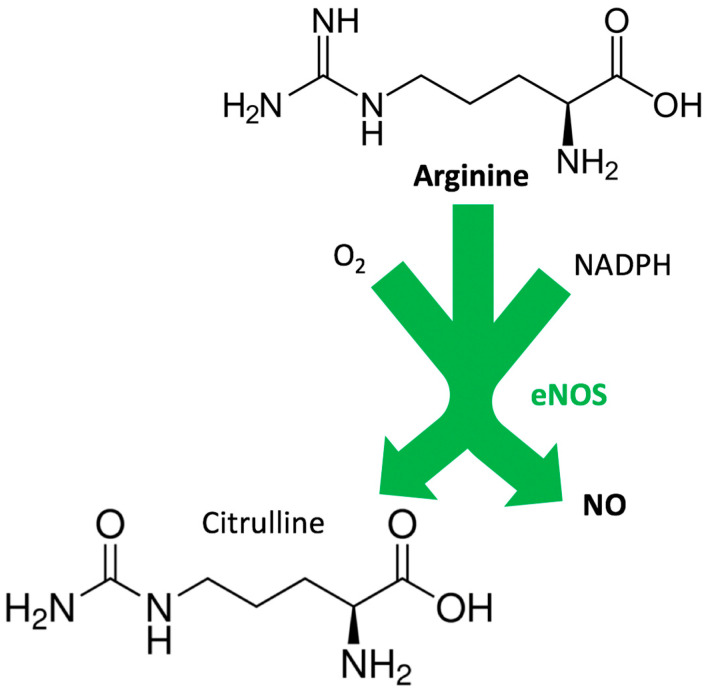
Functional role of arginine in the synthesis of nitric oxide (NO). NADPH: nicotinamide adenine dinucleotide phosphate; eNOS: endothelial NO synthase.

**Figure 2 biomedicines-08-00277-f002:**
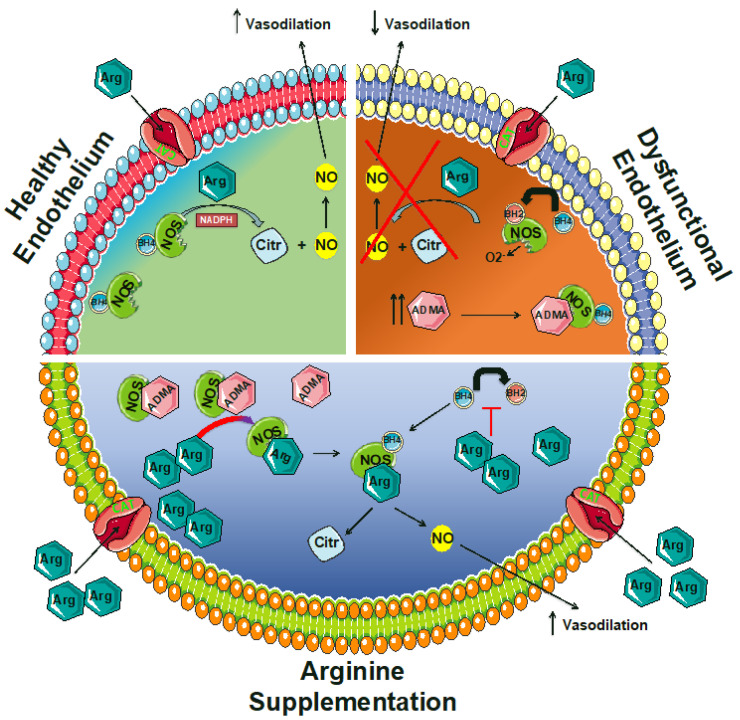
Functional role of arginine in endothelial (dys)function. ADMA: asymmetric dimethylarginine; Arg: arginine; BH2: 7,8-dihydrobiopterin; BH4: tetrahydrobiopterin; CAT: cationic amino acid transporter; Citr: citrulline; NADPH: nicotinamide adenine dinucleotide phosphate; NO: nitric oxide; NOS: NO synthase.

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
