# Peer review of "Arginine and Endothelial Function"

_biomedicines, 2020, doi:10.3390/biomedicines8080277_

Round 1
Reviewer 1 Report
The manuscript by Gambardella and coauthors reviewed the literature related to the impact of Arginine on the endothelial function in animal and human studies. Specifically, they focused on hypertension, CAD, Heart Failure, atherosclerosis.
1. The study failed to address in detail the mechanistic data related to the actions of arginine. The authors discussed only limited number of studies related to mechanisms of action of arginine. Importantly, there was not mention of the phenomenon of ‘eNOS uncoupling’ and the role of arginine. There was not even mention of how arginine and eNOS related in terms of structure of eNOS enzyme. 2. This reviewer do not see why vitamin C is compared to arginine as alternative. The review is about arginine and not about antioxidants in general. 3. There is no mention of the studies of the role of arginine in aging induced endothelial dysfunction. 4. Specific comments are included in the attached file.Author Response
The manuscript by Gambardella and coauthors reviewed the literature related to the impact of Arginine on the endothelial function in animal and human studies. Specifically, they focused on hypertension, CAD, Heart Failure, atherosclerosis.
We thank this Reviewer for the time spent in reviewing our manuscript and for the helpful suggestions.
- The study failed to address in detail the mechanistic data related to the actions of arginine. The authors discussed only limited number of studies related to mechanisms of action of arginine. Importantly, there was not mention of the phenomenon of ‘eNOS uncoupling’ and the role of arginine. There was not even mention of how arginine and eNOS related in terms of structure of eNOS enzyme.
We have expanded these sections, as requested. The phenomenon of ‘eNOS uncoupling’ is now better discussed.
- This reviewer do not see why vitamin C is compared to arginine as alternative. The review is about arginine and not about antioxidants in general.
The section on vitamin C has been removed, as requested.
- There is no mention of the studies of the role of arginine in aging induced endothelial dysfunction.
We now describe the role of arginine in aging induced endothelial dysfunction.
- Specific comments are included in the attached file.
The requested corrections have been done.
Reviewer 2 Report
The review ‘Arginine and Endothelial Function’ by Gambardella J et. al. sounds interesting. In the review the authors indicated arginine, as substrate of NOS, can be converted to NO and discussed supplementation of arginine increases NO bioavailability, in turns altered endothelial dysfunction and potential positive effects on the treatment of hypertension, ischemic heart disease and peripheral artery disease as well as diabetes mellitus.
The review indicates arginine is the subtract of NOS and arginase. Arginine is converted to NO by NOS and improve EC function. However, authors do not discuss the function of arginase in regulation NO bioavailability. Arginase is competing subtract with NOS and in turn regulates NOS function and NO production. It cannot be ignored that the activity of arginase is highly increased in hypertension, ischemic heart disease and peripheral artery disease as well as diabetes mellitus. The positive effect of arginine supplement on the diseases treatment is still unclear, especially in clinic trials, which cannot be excluded from the increased arginase activity in the patients.
The authors discussed the effect vitamin C (VC) on EC function. VC is an important scavenger of reactive oxygen spices ROS). By reduction of ROS, eNOS is re-coupled from uncoupling condition and NO bioavailability is increased in the diseases. There are so many ROS scavenger available. The authors need to explain why VC is so important and emphasized.
The disease mechanism of Covid-19 is still unclear. Although the therapeutic research of Covid-19 is a hot topic, it is not suitable that authors discuss the therapeutic role of arginine in conclusion and perspective part, which may lead to misunderstanding.
The title should be more focused.
Author Response
The review ‘Arginine and Endothelial Function’ by Gambardella J et. al. sounds interesting. In the review the authors indicated arginine, as substrate of NOS, can be converted to NO and discussed supplementation of arginine increases NO bioavailability, in turns altered endothelial dysfunction and potential positive effects on the treatment of hypertension, ischemic heart disease and peripheral artery disease as well as diabetes mellitus.
We thank this Reviewer for the time spent in reviewing our manuscript and for the helpful suggestions.
The review indicates arginine is the subtract of NOS and arginase. Arginine is converted to NO by NOS and improve EC function. However, authors do not discuss the function of arginase in regulation NO bioavailability. Arginase is competing subtract with NOS and in turn regulates NOS function and NO production. It cannot be ignored that the activity of arginase is highly increased in hypertension, ischemic heart disease and peripheral artery disease as well as diabetes mellitus. The positive effect of arginine supplement on the diseases treatment is still unclear, especially in clinic trials, which cannot be excluded from the increased arginase activity in the patients.
Thanks. We now added a paragraph to discuss arginase.
The authors discussed the effect vitamin C (VC) on EC function. VC is an important scavenger of reactive oxygen spices ROS). By reduction of ROS, eNOS is re-coupled from uncoupling condition and NO bioavailability is increased in the diseases. There are so many ROS scavenger available. The authors need to explain why VC is so important and emphasized.
As requested by Reviewer #1, the section on Vitamin C has been removed.
The disease mechanism of Covid-19 is still unclear. Although the therapeutic research of Covid-19 is a hot topic, it is not suitable that authors discuss the therapeutic role of arginine in conclusion and perspective part, which may lead to misunderstanding.
We removed the section on arginine and COVID from the conclusion and perspective.
The title should be more focused.
The title has been changed, as requested.
Round 2
Reviewer 2 Report
I do not have any further comment.
Author Response
I do not have any further comment.
R: Thanks